# Studies of the Variability of Polyphenols and Carotenoids in Different Methods Fermented Organic Leaves of Willowherb (*Chamerion angustifolium* (L.) Holub)

**Marius Lasinskas** [1,*], **Elvyra Jariene** [1], **Nijole Vaitkeviciene** [1], **Jurgita Kulaitiene** [1], **Katarzyna Najman** [2] **and Ewelina Hallmann** [2]

[1] Institute of Agriculture and Food Sciences, Agriculture Academy, Vytautas Magnus University, Donelaicio str. 58, 44248 Kaunas, Lithuania; elvyra.jariene@vdu.lt (E.J.); nijole.vaitkeviciene@vdu.lt (N.V.); jurgita.kulaitiene@vdu.lt (J.K.)

[2] Department of Functional and Organic Food, Institute of Human Nutrition Sciences, Warsaw University of Life Sciences, Nowoursynowska 15c, 02-776 Warsaw, Poland; katarzyna_najman@sggw.edu.pl (K.N.); ewelina_hallmann@sggw.edu.pl (E.H.)

\* Correspondence: marius.lasinskas@vdu.lt; Tel.: +370-676-82266

**Abstract:** The demand for organic production is increasing worldwide. The willowherb, grown in an organic way, contributes greatly to the idea of a healthier society and clean land. Willowherb is widespread in the world and has high polyphenols, carotenoids, and antioxidant properties. The purpose of this work was to investigate the influence of solid-phase fermentation (SPF) under different conditions on the variation of polyphenols and carotenoids in the organic leaves of willowherb. The leaves were fermented for different periods of time: 24, 48, and 72 h; and in aerobic and anaerobic conditions. The evaluation of polyphenols and carotenoids was completed using high-performance liquid chromatography (HPLC), and antioxidant activity was measured with spectrophotometric method. Hierarchical cluster analysis was used to describe differences in biologically active compounds between willowherb samples. The experiment showed that the highest quantities of total phenolic acids and flavonoids were determined after 24 h under aerobic SPF, but the amountof total carotenoids was higher after 72 h anaerobic SPF, compared to control. Not-fermented willowherb leaves had a lower antioxidant activity. compared to fermented leaves. In conclusion, SPF can be used to change polyphenol and carotenoid quantities in organic leaves of willowherb.

**Keywords:** organic willowherb; solid-phase fermentation; polyphenols; carotenoids; antioxidant activity

## 1. Introduction

At present, foods and food supplements rich in antioxidants are very important, but more important is that these products are made from organic raw materials. More and more people tend to consume organic products, because they are associated with higher quality, compared to conventionally grown products. The willowherb (*Chamerion angustifolium* (L.) Holub) (also known as fireweed) is a plant that can solve health problems, and is a very popular medicinal plant in Lithuania. It grows in various soils, but usually in damaged areas: burned or cut forests and highways. It can grow in light forests, but not in full shade [1]. *Chamerion angustifolium* is a plant used as a medicine worldwide, in traditional medicines for the treatment of various disorders. Studies have shown thatwillowherbextracts have a wide range of therapeutic effects: anti-proliferative, anti-inflammatory, antibacterial, and antioxidant [2]. Prasad et al. described positive effects of willowherb leaves tea

in a variety of diseases: anemia, headaches, infections, colds, gastrointestinal, prostate, and urinary problems [3]. Finding a method to enhance the bioavailability of willowherb polyphenols and carotenoids is a very important task [4]. High levels of bioactive compounds in plants are very important, but their availability during infusion is also essential [5]. Thus, it is very important to study the chemical component variations in not-fermented and fermented leaves of willowherb. It is also very important to find a way to increase the efficient absorption of bioactive compounds from leaves. One of the current ways to improve extraction and modify bioactive compounds in willowherb fermented leaves and their bioavailability is solid-phase fermentation (SPF). It is expectedthat cutting and pressing during SPF could intensify cell wall degradation, thus improving diffusion of bioactive compounds from the inner cell parts, and then initiating better extraction. In addition, there is evidence that some solid-phase fermentation parameters may activate the accumulation process of some bioactive compounds in willowherb leaves [6,7].

Willowherbs are perfect source of polyphenols, carotenoids, and other compounds. In their leaves, many phenolic acids and flavonoids have been found: benzoic, ellagic and gallic acids, myricetin, quercetin, quercetin-3-*O*-rutinoside, luteolin, quercetin-3-*O*-glucoside, kaempferol, tannin oenothein B, and carotenoids: lutein, zeaxanthin, beta-carotene [8–10].

On the other hand, there are not enough researches on the influence of different SPF conditions (duration, aerobic, and anaerobic process) on the chemical components of organic willowherb leaves. The purpose of this experiment was to determine the influence of SPF under various conditions on the quantities of polyphenols (phenolic acids, tannin oenothein B, flavonoids) and carotenoids (lutein, zeaxanthin, beta-carotene) in organic willowherb leaves. According to the results of this experiment, SPF with various conditions can be used to prepare high-quality organic products (foods and food supplements) from organic willowherb leaves.

## 2. Materials and Methods

### 2.1. Origin of Material

The organic leaves of willowherb (*Chamerion angustifolium* (L.) Holub) that grew up in Jonava district, Safarkos village, Giedres Nacevicienes organic farm (No. SER-T-19-00910, Lithuania) were picked and investigated in 2019.

### 2.2. Plant Material Preparation

The organic leaves ofwillowherb (*Chamerion angustifolium* (L.) Holub) were gathered at random from various places of the plot at the outset of mass blossom in July. The composite leaf sample was 6.3 kg. For experiments in the laboratory, the samples were divided into several components:

1. Control: 0.900 kg notfermented (0 h).
2. Aerobic: 2.7 kg for SPF lasting 24, 48, and 72 h.
3. Anaerobic: 2.7 kg for SPF lasting 24, 48, and 72 h (Table 1).

**Table 1.** Laboratory experiment variants.

| Fermentation Duration | | | |
|---|---|---|---|
| **0 h** | **24 h** | **48 h** | **72 h** |
| 0.900 kg control (notfermented) | 0.900 kg aerobic SPF [1]<br>0.900 kg anaerobic SPF | 0.900 kg aerobic SPF<br>0.900 kg anaerobic SPF | 0.900 kg aerobic SPF<br>0.900 kg anaerobic SPF |

[1] SPF—solid-phase fermentation.

During the SPF, to cut fresh leaves, special plastic knives were used. The samples were divided into parts of 0.300 kg. For anaerobic SPF: the leaves were thoroughly pressed into containers (made from glass) and covered with a lid. For aerobic SPF: glass containers were covered with an air-passing lid. The SPF passed at a temperature of 30 °C in a dark chamber for durations: 24, 48, and 72 h.

All variations of the experiment were carried out in three replications. Then, the raw leaves were lyophilized in freeze-drying plant sublimator (ZIRBUS GmbH, Harz, Germany), and leaves were milled and kept in closed containers at a temperature of 25 °C in the ventilated, dry, dark, and cool room. All chemical experimentations were done three times.

## 2.3. Measurement of Polyphenols

Ponder and Hallmann (2019) described the method for polyphenols determination using high-performance liquid chromatography (HPLC) [11]. A total of 100 mg of powder of freeze-dried willowherb leaves was mixed up with 5 mL of 80% methanol (80:20 methanol and ultrapure water), and then plastic tubes were closed by plastic cap and shaken with vortex (60 s). Then, in an ultrasonic bath, all examples were extracted for 10 min, ata temperature of 30 °C, and 5.5 kHz. After 15 min, the examples were centrifuged for 10 min, 3780× *g*, and temperature was 5 °C. A clean plastic tube was used to collect the supernatant, and then it was centrifuged once more for 5 min, 31,180× *g*, at 0 °C temperature. Supernatant (850 µL) was moved to a vial (HPLC) and was analyzed. For polyphenols separation, a Synergi Fusion-RP 80i Phenomenex column (250 × 4.60 mm) was used. Shimadzu equipment (two pumps (LC-20AD), controller (CBM-20A), column oven (SIL-20AC), and spectrometer UV/Vis (SPD-20 AV)) was used to carry out the analysis. The phenolics were isolated with these gradient conditions: a flow rate of 1 mL min$^{-1}$; we used two gradient phases: 10% (*v/v*) acetonitrile and ultrapure water (phase A) and 55% (*v/v*) acetonitrile and ultrapure water (phase B). Orthophosphoric acid was used to acidify the phases (pH 3.0). The whole analysis lasted 38 min. The program of the stage was: 1.00–22.99 min, 95% phase A and 5% phase B; 23.00–27.99 min, 50% phase A and 50% phase B; 28.00–28.99 min, 80% phase A and 20% phase B; and 29.00–38.00 min, 95% phase A and 5% phase B. For flavonoids, wavelengths were 250 nm, and for phenolic acids, 370 nm. The pure standards 99.9% (Poland, Sigma-Aldrich) were used to identify the phenolics.

## 2.4. Measurement of Carotenoids

Hallmann et al. described the method for carotenoids determination [12]. A total of 100 mg of powder from freeze-dried leaf sample was added into a plastic test tube, and then 1 mg magnesium carbonate and 5 mL of pure acetone were added. In next step, the solution was mixed carefully using vortexing, and an ultrasonic bath was used for incubation for 10 min at 0 °C temperature. Then the examples were centrifuged (10 min, 3780× *g*, 5 °C). HPLC was used to take an aliquot of 800 µL of extract for testing. For carotenoids determination, HPLC (Shimadzu, USA Manufacturing Inc., Canby, OR, USA) was used, consisting of two LC-20AD pumps, an SIL-20AC autosampler, a CMB-20A system controller, an ultraviolet–visible detector SPD-20AV, and an oven CTD-20AC. A Synergi Max-RP 80i column 250 × 4.60 mm (Phenomenex Inc., Torrance, CA, USA) was used, and then elution was done with gradient flow in two mobile phases: (A) acetonitrile/methanol (90:10), and (B) methanol/ethyl acetate (64:36). The used gradient program was: 0–16 min—95% Solvent A and 5% Solvent B; 17–21 min—50% Solvent A and 50% Solvent B; 22–25 min—20% Solvent A and 80% Solvent B; 26–28 min—20% Solvent A and 80% Solvent B. The whole analysis lasted 28 min, the flow rate was 1 mL min$^{-1}$, and the wavelength range for detection was 445–450 nm. The external standards (Fluka and Sigma-Aldrich, Poznan, Poland) were used for identification of carotenoids.

## 2.5. Antioxidant Activity

Srednicka-Tober et al. described the method for antioxidant activity determination [13]. The sample of 250 mg freeze-dried plant powder was placed into a plastic tube, and distilled water was added (25 mL). The vortex was used for 1 min to mix the samples (Labo Plus, Warsaw, Poland). In the next stage, the samples were placed into a shaker incubator (IKA, Staufen im Breisgau, Germany) for 1 h, at 30 °C. Then, the samples were shaken again and centrifuged (Centrifuge, MPW-380 R, Warsaw, Poland) for 15 min, 14,560× *g*, at temperature 5 °C. After that, the supernatant was gathered for measurements. In laboratory glass tubes, the samples were determined with a measurement dilution

scheme (0.5–1.5 mL) and then placed in 3.0 mL of ABTS·+ cationic solution in phosphate-buffered saline (PBS). After 6 min, a spectrophotometer (Helios $\gamma$, Thermo Scientific, Warsaw, Poland) was used to take the samples' absorbances (21 °C, wavelength $\lambda = 734$ nm). The received calculations were measured with a special formula, also the dilution factor was used. The conclusive data were expressed in mmol of TE (Trolox equivalents/100 g dry weight (DW)).

*2.6. Statistical and Multivariate Analysis*

The ANOVA (a two-way analysis of variance) method, with software package STATISTICA, was used to perform statistical analysis of obtained data (Statistica 12; StatSoft, Inc., Tulsa, OK, USA). The results were submitted as the mean with standard error. The Fisher's LSD test was used to estimate the statistical significance of differences between the means ($p < 0.05$). To characterize differences in biologically active substances (polyphenols and carotenoids) between willowherb leaves samples, the hierarchical cluster analysis was used (XLSTAT Software, XLSTAT, 2016, New York, NY, USA).

## 3. Results

*3.1. Content of Bioactive Compounds*

One of the ways to improve bioactive compound extraction from willowherb leaves is to use SPF in various conditions. In this experiment, SPF changed the amounts of polyphenols and carotenoids in organic fermented willowherb leaves under aerobic and anaerobic conditions. The content of total phenolic acids, polyphenols, and flavonoids was highest under the aerobic fermentation conditions, but the total quantity of carotenoids was higher in anaerobic fermentation (Table 2). The amount of individual bioactive compounds differed according to fermentation conditions.

According to Schepetkin et al. (2016), ellagitannins, such as oenothein B, are among the compounds considered to be the primary biologically active components in willowherb extracts [2]. The quantity of tannin oenothein B in aerobic fermentation (24 h) was 19.25% higher compared to control (not-fermented), and 53.88% higher than under anaerobic conditions (24 h) (Table 3). In willowherb leaves fermented for 48 h, oenothein B quantity decreased by 13.07%, and in fermented leaves for 72 h—33.08%.

Flavonoids were found as the prime group of naturally-occurring phenolic compounds [14]. It was established that the higher amount of flavonoid quercetin-3-*O*-rutinoside was determined in not-fermented willowherb leaves (control), and after 48 h under the anaerobic fermentation. The highest quantities of myricetin, luteolin, quercetin, and kaempferol were determined after fermentation in aerobic conditions, but the highest amount of quercetin-3-*O*-glucoside was identified after 24 h under anaerobic fermentation conditions (Table 3).

Phenolic acids can be used to treat oxidative damage illnesses (cancer and heart diseases), and they are the plant's secondary metabolites [15]. Our results show that the highest amount of gallic and benzoic acids occurred after 72 h anaerobic fermentation. The highest quantity of chlorogenic acid (9.26 mg 100 g$^{-1}$DW) was determined in not-fermented organic willowherb leaves (control), for ellagic acid (1630.68 mg 100 g$^{-1}$DW) after 24 h aerobic fermentation, and *p*-coumaric after 24 h anaerobic fermentation (345.85 mg 100 g$^{-1}$ DW) (Table 4).

Carotenoids are very important for human health, and it is advised that they have to be a regular part of daily food. Carotenoids are found in many plant products [16]. According to our results in willowherb leaves samples, all individual carotenoids levels were significantly higher after 24 and 72 h anaerobic fermentation: lutein 27.83 mg 100 g$^{-1}$ DW (24 h) and 29.75 mg 100 g$^{-1}$ DW (72 h), zeaxanthin 13.96 mg 100 g$^{-1}$ DW (24 h) and 14.32 mg 100 g$^{-1}$ DW (72 h), and beta-carotene 6.31 mg 100 g$^{-1}$ DW (24 h) and 6.30 mg 100 g$^{-1}$ DW (72 h) (Table 5).

**Table 2.** The influence of solid-phase fermentation (SPF) on the matter of bioactive compounds (mg 100 g$^{-1}$ dry weight (DW)) in organic willowherb leaves. Mean value ± standard error, $n = 3$.

| Fermentation Duration/Bioactive Compounds | Total Polyphenols | Total Phenolic Acids | Total Flavonoids | Total Carotenoids |
|---|---|---|---|---|
| Control (notfermented) | 2681.96 ± 62.01 b [1] | 1654.75 ± 80.14 c | 1027.21 ± 102.07 c | 36.34 ± 1.05 e |
| Aerobic fermentation method | | | | |
| Fermented 24 h | 3209.20 ± 152.06 a | 1895.23 ± 59.33 a | 1313.97 ± 99.75 a | 34.80 ± 1.47 e |
| Fermented 48 h | 2772.47 ± 13.21 b | 1615.34 ± 27.69 cd | 1157.13 ± 13.43 b | 39.64 ± 0.12 d |
| Fermented 72 h | 2458.52 ± 95.04 c | 1544.15 ± 40.46 d | 914.37 ± 59.05 d | 34.70 ± 0.44 e |
| Anaerobic fermentation method | | | | |
| Fermented 24 h | 2509.46 ± 59.25 c | 1818.43 ± 62.67 ab | 691.03 ± 10.67 e | 48.10 ± 0.95 b |
| Fermented 48 h | 2282.67 ± 65.54 d | 1572.60 ± 15.90 cd | 710.07 ± 68.90 e | 42.65 ± 2.18 c |
| Fermented 72 h | 2142.77 ± 81.14 d | 1760.46 ±65.80 b | 382.32 ± 3.25 f | 50.38 ± 1.24 a |
| *p*-Value (SPF [2] duration × SPF method) | <0.00011 | <0.00216 | <0.00001 | <0.00001 |

[1] The differences between the means in columns marked by not the same small letter (a, b, c, d, e, f) are significant at $p < 0.05$. [2] SPF—solid-phase fermentation.

**Table 3.** The influence of SPF on the content of flavonoids and tannin (mg 100 g$^{-1}$ DW) in organic willowherb leaves. Mean value ± standard error, $n = 3$.

| Fermentation Duration/Bioactive Compounds | Oenothein B | Quercetin-3-*O*-Rutinoside | Myricetin | Luteolin | Quercetin | Quercetin-3-*O*-Glucoside | Kaempferol |
|---|---|---|---|---|---|---|---|
| Control (not fermented) | 969.07 ± 72.77 b [1] | 23.31 ± 0.86 b | 12.11 ± 0.13 cd | 2.70 ± 0.04 c | 2.45 ± 0.17 e | 15.21 ± 3.73 f | 2.35 ± 0.09 cd |
| Aerobic fermentation method | | | | | | | |
| Fermented 24 h | 1200.02 ± 68.26 a | 20.07 ± 0.02 c | 29.13 ± 0.24 a | 2.95 ± 0.02 b | 10.65 ± 0.12 a | 47.88 ± 1.50 e | 3.27 ± 0.22 b |
| Fermented 48 h | 1043.16 ± 30.39 b | 18.23 ± 0.05 d | 26.97 ± 0.89 b | 3.93 ± 0.19 a | 3.73 ± 0.21 c | 57.31 ± 0.14 d | 3.81 ± 0.13 a |
| Fermented 72 h | 803.07 ± 61.68 c | 22.87 ± 0.03 b | 11.14 ± 1.36 de | 3.74 ± 0.13 a | 5.81 ± 0.26 b | 64.44 ± 3.43 c | 3.30 ± 0.06 b |
| Anaerobic fermentation method | | | | | | | |
| Fermented 24 h | 553.47 ± 37.36 d | 20.86 ± 2.04 c | 12.45 ± 0.26 c | 3.83 ± 0.24 a | 3.92 ± 0.35 c | 94.29 ± 4.91 a | 2.21 ± 0.06 d |
| Fermented 48 h | 582.74 ± 52.05 d | 25.23 ± 3.08 a | 10.59 ± 0.81 ef | 2.37 ± 0.03 d | 2.69 ± 0.19 e | 84.02 ± 0.67 b | 2.43 ± 0.04 c |
| Fermented 72 h | 330.56 ±5.18 e | 19.54 ± 0.82 cd | 9.42 ± 0.75 f | 2.43 ± 0.09 d | 3.11 ± 0.09 d | 14.89 ± 0.72 f | 2.37 ± 0.01 cd |
| *p*-Value (SPF [2] duration × SPF method) | <0.00001 | <0.00001 | <0.00001 | <0.00001 | <0.00001 | <0.00001 | <0.00001 |

[1] The differences between the means in columns marked by not the same small letter (a, b, c, d, e, f) are significant at $p < 0.05$. [2] SPF—solid-phase fermentation.

**Table 4.** The influence of SPF on the content of phenolic acids (mg 100 g$^{-1}$ DW) in organic willowherb leaves. Mean value ± standard error, *n* = 3.

| Fermentation Duration/Bioactive Compounds | Gallic | Chlorogenic | *p*-Coumaric | Ellagic | Benzoic |
|---|---|---|---|---|---|
| Control (not fermented) | 6.59 ± 0.40 c [1] | 9.26 ±0.23 a | 121.08 ± 5.47 e | 1507.40 ± 73.92 b | 10.42 ± 1.72 b |
| Aerobic fermentation method | | | | | |
| Fermented 24 h | 13.57 ± 0.38 b | 3.17 ± 0.08 e | 237.11 ± 7.30 c | 1630.68 ± 57.14 a | 10.70 ± 0.97 b |
| Fermented 48 h | 4.47 ± 1.01 d | 9.05 ± 0.15 a | 120.67 ± 10.26 e | 1477.35 ± 19.60 b | 3.81 ± 1.13 c |
| Fermented 72 h | 5.01 ± 0.30 d | 6.17 ± 0.29 c | 165.92 ± 12.33 d | 1362.84 ± 29.15 c | 4.22 ± 0.81 c |
| Anaerobic fermentation method | | | | | |
| Fermented 24 h | 4.68 ± 0.54 d | 7.19 ± 0.71 b | 345.85 ± 20.03 a | 1445.82 ± 45.08 bc | 14.89 ± 1.95 a |
| Fermented 48 h | 15.77 ± 0.37 a | 4.37 ± 0.05 d | 290.61 ± 8.64 b | 1249.77 ± 16.74 d | 12.07 ± 0.50 b |
| Fermented 72 h | 16.42 ± 0.92 a | 3.58 ± 0.03 e | 283.76 ± 0.88 b | 1441.32 ± 158.83 bc | 15.37 ± 0.08 a |
| *p*-Value (SPF [2] duration × SPF method) | <0.00001 | <0.00001 | <0.00001 | <0.00027 | <0.00001 |

[1] The differences between the means in columns marked by not the same small letter (a, b, c, d, e) are significant at *p* < 0.05. [2] SPF—solid-phase fermentation.

**Table 5.** The influence of SPF on the content of carotenoids (mg 100 g$^{-1}$ DW) in organic willowherb leaves. Mean value ± standard error, *n* = 3.

| Fermentation Duration/Bioactive Compounds | Lutein | Zeaxanthin | Beta-Carotene |
|---|---|---|---|
| Control (not fermented) | 19.08 ± 1.04 d [1] | 11.16 ± 0.06 c | 6.10 ± 0.14 b |
| Aerobic fermentation method | | | |
| Fermented 24 h | 18.16 ± 1.49 d | 11.57 ± 0.06 c | 5.07 ± 0.02 d |
| Fermented 48 h | 22.30 ± 0.07 c | 12.27 ± 0.06 b | 5.08 ± 0.02 d |
| Fermented 72 h | 18.50 ± 0.53 d | 12.12 ± 0.17 b | 4.09 ± 0.13 e |
| Anaerobic fermentation method | | | |
| Fermented 24 h | 27.83 ± 0.63 a | 13.96 ± 0.39 a | 6.31 ± 0.07 a |
| Fermented 48 h | 24.61 ± 2.15 b | 12.36 ± 0.41 b | 5.68 ±0.03 c |
| Fermented 72 h | 29.75 ± 0.87 a | 14.32 ± 0.51 a | 6.30 ± 0.07 a |
| *p*-Value(SPF [2] duration × SPF method) | <0.00001 | <0.00001 | <0.00001 |

[1] The differences between the means in columns marked by not the same small letter (a, b, c, d, e) are significant at *p* < 0.05. [2] SPF—solid-phase fermentation.

### 3.2. Antioxidant Activity in Willowherb Leaves

We noticed that not-fermented willowherb leaves (control) had an inferior antioxidant activity, in comparison with 24 h, 48 h, and 72 h SPF. In anaerobic conditions, samples had a higher antioxidant activity in proportion to aerobic SPF. The short duration (24 h) under the aerobic fermentation gave higher results of antioxidant activity of willowherb leaves, compared to those under the short duration (24 h) in anaerobic fermentation (Figure 1).

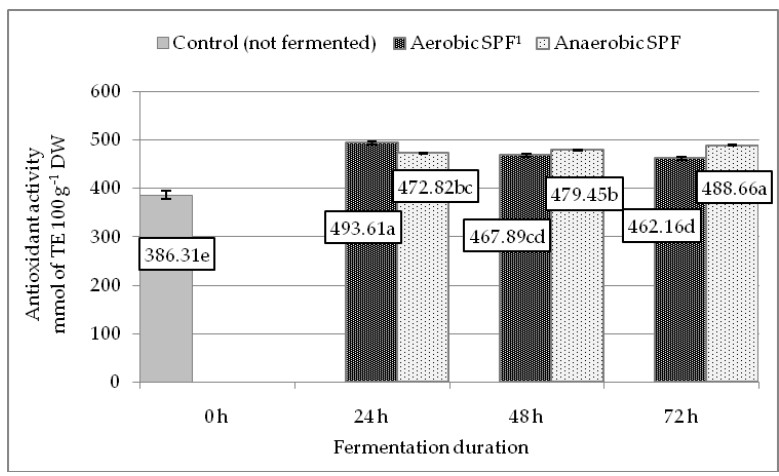

**Figure 1.** The influence of SPF on the antioxidant activity of organic willowherb leaves ($p < 0.0001$). Means followed by the same letter are not significantly different ($p < 0.05$), $n = 3$. [1] SPF—solid-phase fermentation.

### 3.3. Allocation of Bioactive Compounds in Willowherb Leaves Samples

In this research, hierarchical cluster analysis was performed to group willowherb samples fermented at different conditions. The dendrogram received from hierarchical cluster analysis is shown in Figure 2. The clustering of sample types regarding the composition of bioactive compounds and antioxidant activity grouped them into three different groups (C1, C2, and C3). Therefore, this result shows that the three clusters have different chemical compositions in terms of bioactive compounds and antioxidant activity.

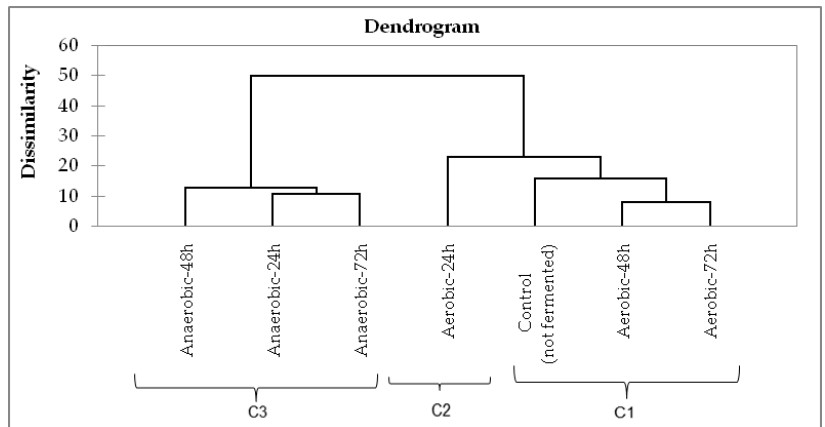

**Figure 2.** Cluster dendrogram for organic willowherb samples fermented under the different solid-phase fermentation method and different duration, based on 20 variables. These samples are grouped using Ward's method and Euclidean distance. Conditions of solid-phase fermentation: different fermentation method (Control (not fermented), aerobic solid-phase fermentation method (Aerobic) and anaerobic solid-phase fermentation method (Anaerobic)) and different fermentation duration (Control (not fermented), fermented 24 h, 48 h, and 72 h).

Cluster 1 demonstrated that control (not fermented), aerobic-48 h, and aerobic-72 h samples were grouped closely due to higher contents of luteolin and chlorogenic acid (Figure 2). Aerobic-24 h formed cluster 2 (C2) in the dendrogram. This group was formed regarding the higher contents of total polyphenols, total flavonoids, oenothein B, myricetin, quercetin, and ellagic acid. Additionally, anaerobic-24 h, anaerobic-48 h, and anaerobic-72 h samples were grouped into cluster 3 (C3), due to their stronger antioxidant activity and higher contents of phenolic acids such as gallic, *p*-coumaric, and benzoic, as well as total carotenoids and individual carotenoids (lutein, zeaxanthin, and beta-carotene).

## 4. Discussion

Phenolic compound and carotenoid synthesisisactivated in response to climatic, humidity, or other stressful impacts on the plant. 2019 was too dry for the development of willowherb during their vegetative period, so it could have been one of the key factors in monitoring higher levels of polyphenols and carotenoids.

Microbial metabolism and enzymes produced during SPF, such as polyphenol oxidase, have a significant effect on the components of fermented willowherb leaves, by breaking down macromolecular components (proteins, lipids, and polysaccharides) into lower molecular weight substances and secondary products of metabolism [17]. Also, cutting and pressing during SPF could intensify cell wall degradation, thus improving diffusion of bioactive compounds from the inner cell parts, and then initiating better extraction. Thus, it could have been one of the key factors in monitoring higher levels of polyphenols and carotenoids after solid-phase fermentation.

In this experiment, we determined that the short-term (24 h) aerobic solid-phase fermentation process increased the level of total polyphenolic compounds by 19.66%. The amount of total phenolic acids significantly increased (by 14.53%) after 24 h aerobic fermentation, and total flavonoids increased by 27.92%, compared to control. Contrarily, the highest amounts of total carotenoids were observed after anaerobic fermentation: 50.38 mg 100 $g^{-1}$ DW (72 h) and 48.10 mg 100 $g^{-1}$ DW (24 h).

Ellagitannin dimer oenothein B quantity depends on ellagic and gallic acid quantities. The amount of oenothein B significantly increased after 24 h aerobic SPF, and significantly decreased after anaerobic SPF (Table 3). This can be explained by the higher oxygen amount and the changes in quantity and quality of gallic and ellagic acids during aerobic solid-phase fermentation. Also, microbial metabolism intensifies and the enzyme (polyphenol oxidase) activity disintegrates them into lower molecular weight components [18].

Quercetin and quercetin-3-*O*-rutinoside (rutin) are very biochemically and structurally related to each other, thus we can explain the results among flavonoid quantities (Table 3). Quercetin significantly increased after 24 h aerobic fermentation, but the amount of quercetin-3-*O*-rutinoside significantly decreased. It can be explained by the disintegration of glycoside links by fermentation bacteria [19].

Myricetin, luteolin, and quercetin are structurally similar. Myricetin can be produced from kaempferol [20]. The levels of myricetin and kaempferol were significantly higher after aerobic fermentation (Table 3).

The highest amounts of gallic acid were determined after 24 h aerobic, and anaerobic fermentation after 48 h and 72 h. Gallic acid is found as part of hydrolysable tannins or in free form. Gallic acid and ellagic acid groups are related. The results could be explained, because hydrolysable tannins (gallotannins and ellagitannins) are disintegrating during hydrolysis into gallic acid and glucose [21].

Cinnamic acid takes part in the metabolism of chlorogenic and *p*-coumaric acids [22]. Thus, we can explain the results of these two phenolic acids. Chlorogenic acid levels decreased significantly in almost all variants of fermentation, but *p*-coumaric acid increased significantly only after anaerobic solid-phase fermentation (Table 4).

The most common phenolic acid found in willowherb leaves is ellagic acid. In plants, ellagitannins hydrolyze into ellagic acids [23]. In this case, SPF significantly increased the level of ellagic acid only after 24 h aerobic fermentation, and decreased it in all other variants (Table 4). During SPF, microbial

metabolism and enzyme activity rises, and part of the ellagic acid disintegrates to the final acids. Due to this fact, the amounts of ellagic acid that decreased could be clarified.

Benzoic acid is found in a lot of plants and it mediates the biosynthesis of many secondary metabolites [24]. The quantities of benzoic acid in willowherb were significantly raised only under anaerobic SPF (Table 4). These changes in benzoic acid amounts could be explained, because anaerobic bacterial metabolism and active enzyme activity lead the composition of benzoic acid from macromolecular substances.

Lutein is isomeric with zeaxanthin(both belong to the xanthophylls group). Lutein and zeaxanthin can be interconverted in the body through an intermediate called meso-zeaxanthin [25]. Beta-carotene (carotenes group) is hydrocarbon, and contains no oxygen. All these carotenoids areformationsoftetraterpenes (they are produced from 8isoprenemolecules and contain 40 carbon atoms).

The significantly higher amounts of lutein, zeaxanthin, and beta-carotene were determined after anaerobic fermentation, compared to control and aerobic SPF (Table 5). The carotenoid precursor, geranyl geranyl diphosphate (GGPP), can be converted into carotenes or xanthophylls by undergoing different steps within the carotenoid biosynthetic pathway [26]. The last two steps, involving (e)-4-hydroxy-3-methylbut-2-en-1-yl diphosphate (HMBPD) synthase and reductase, can only occur in completely anaerobic environments [27]. Thus, we can explain, why higher amounts of lutein, zeaxanthin, and beta-carotene were determined in anaerobic conditions.

Couto et al. suggest that the final quality of a solid-phase fermentation process can be influenced not only by the duration of the fermentation, but also by other factors, such as certain microorganisms: bacteria, yeast, fungi, and others [28].

## 5. Conclusions

The results show that solid-phase fermentation significantly improves the isolation of bioactive compounds from the leaves of willowherb. Moreover, the selected organic willowherb samples varied significantly in the composition of polyphenolic compounds and carotenoids. The highest amounts of total phenolic acids and total flavonoids were determined after 24 h under aerobic solid-phase fermentation, but the amountof total carotenoids was higher after 72 h using anaerobic solid-phase fermentation, compared to control (not fermented). Not-fermented willowherb leaves had a lower antioxidant activity, compared to fermented leaves.

In conclusion, SPF could intensify cell wall degradation, thus improving better diffusion of bioactive compounds from the inner cell parts, and then initiating better extraction. Thus, it could be one of the key factors in monitoring higher levels of polyphenols and carotenoids after solid-phase fermentation. According to experimental results, 24 h or 48 h duration aerobic solid-phase fermentation could be recommended for organic willowherb health-promoting products. It needs to be considered that the proposed method and different conditions will give the assumed results with organic willowherb (*Chamerion angustifolium* (L.) Holub) leaves or plants with such composition.

**Author Contributions:** Conceptualization: M.L.; data curation: M.L.; formal analysis: N.V.; funding acquisition: E.J.; methodology: K.N., E.H., and J.K.; project administration: E.J.; resources: M.L.; software: N.V.; supervision: E.J. and J.K.; validation: E.H.; visualization: M.L. and E.H.; writing—original draft: M.L.; writing—review and editing: M.L. and E.J. All authors have read and agreed to the published version of the manuscript.

**Funding:** This research received no external funding.

**Conflicts of Interest:** The authors declare no conflict of interest.

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
