# Peer review of "Studies of the Variability of Polyphenols and Carotenoids in Different Methods Fermented Organic Leaves of Willowherb (Chamerion angustifolium (L.) Holub)"

_applsci, doi:10.3390/app10155254_

Round 1

Reviewer 1 Report

Dear authors,

Below you can find comments and suggestions

  • Originality/Novelty:  The research is interesting, but it has several disadvantages. The question is- how it is possible to performed effective fermentation in willowherb, since the leaf of most species contains a small proportion of sugar (especially in 24 hours)? It would be useful to state the amount of sugar in the willowherbs.

Also, fermentation occurs as a result of the action of various microorganisms, and in the paper such data are not mentioned, e.g. their possible influence on the fermentation flow, increasing or decreasing polyphenols or carotenoids etc. There are many factors which can influence on the efficiency of the extraction targeted bioactive compounds.

  • Significance:  The results are interpreted appropriately, but some additional data could be improved the manuscript quality (see other comments).
  • Quality of Presentation:  The summary highlights the importance of polyphenols but not carotenoids in willowherb, although the influence of SFF on these bioactive compounds also has been investigated.

Line 22-23: The evaluation of polyphenols, carotenoids and antioxidant activity in the organic  leaves was completed using high-performance liquid chromatography (HPLC). --- This sentence is not entirely correct since AOA was not determined by HPLC.

In the introduction, the focus should be more on the applied SFF process and the influence of SFF on the formation of metabolites (of the phenols or carotenoids), etc. This part is not sufficiently described and highlighted, despite the fact that the whole work is based on it. Furthermore, nowhere is it pointed out which processes or compounds during SFF affect the increase in total or individual compounds. It would be realistic to expect cell wall degradation and better diffusion of target compounds from the inner cell parts, improved extraction, etc . Target bioactive compounds are unlikely to accumulate during SFF. The same comment can be applied in the part of the discussion of the obtained results.

Line 145-146. One of the ways to change bioactive compounds in willowherb is to use SFF in various conditions. …….The mentioned statement does not describe the actual results. According to my opinion, SFF has more of an effect on improving the extraction yield of phenolic compounds from willowherb than on altering bioactive compounds. The change would be, if the certain metabolites that are not present in fresh leaves, are formed during SFF.

In the discussion, the authors initially emphasize the influence of the climatic conditions (higher daily temperature etc.) on the accumulation of bioactive compounds, polyphenols and carotenoids. Later, these influences are nowhere mentioned nor they have been the subject of performed research. Therefore, the importance of the SFF should be highlighted at the beginning of the discussion.

(Phenolic compounds and carotenoids synthesis is activated in response to climatic, humidity or other stressful impacts on the plant. 2019 was too dry for the development of willowherb during their vegetative period, so it could have been one of the key factors in monitoring higher levels of polyphenols and carotenoids. ------ This sentence would make a sense if you had the data for previously years.)

Line 228-231: This impact also was not researched, but you can use this fact if you want to explain some phenomena during SFF.

The discussion is rather poorly written. The authors should consider in explaining obtained results, the influence of SFF, due to there is a little possibility of myricetin formation from kaempferol, after the plant has been harvested and exposed to some process.

  • Interest to the Readers:  In conclusion, the efficacy of SFF on the isolation of polyphenols and carotenoids should be emphasized more. Also, repetition of the discussion should be avoided in highlighting the conclusions.
  • English Level: Is the English language appropriate and understandable?

English is understandable, but requires a revision of the native speaker.

Reviewer 2 Report

„The recieved calculations were measured with a special formula, also the dilution factor was used. The conclusive data were expressed in mmol of TE (Trolox equivalents/100 g DW).” - Please specify the model which was used? Has a calibration curve been made for Trolox? In what concentrations?

Have the authors investigated the microbial composition of fermented products? There are no references to specific types of microorganisms in the discussion. Please complete the literature if the authors did not determine it.

Reviewer 3 Report

Presented for grading manuscript : Studies of the variability of polyphenols and carotenoids in different methods ferm ented organic leaves of willowherb (Chamerion angustifolium (L.)  Holub) concerns the determination of antioxidant properties expressed as overall content of polyphenols and karotenoids w willowherb. In general the manuscript is prepared correctly, analitical models are precisely described, results are presented in tables and on charts. Suggestions are posted in comments

Round 2

Reviewer 1 Report

The manuscript has been improved and all suggestion were accepted.

Carotenoids should be added as keyword.